# Lymph Node Molecular Analysis with OSNA Enables the Identification of pT1 CRC Patients at Risk of Recurrence: A Multicentre Study

**DOI:** 10.3390/cancers15225481

**Published:** 2023-11-20

**Authors:** Karmele Saez de Gordoa, Maria Teresa Rodrigo-Calvo, Ivan Archilla, Sandra Lopez-Prades, Alba Diaz, Jordi Tarragona, Isidro Machado, Juan Ruiz Martín, Diana Zaffalon, Maria Daca-Alvarez, Maria Pellisé, Jordi Camps, Miriam Cuatrecasas

**Affiliations:** 1Pathology Department, Centre of Biomedical Diagnosis (CDB), Hospital Clinic, 08036 Barcelona, Spain; saezdegord@clinic.cat (K.S.d.G.); mtrodrigo@clinic.cat (M.T.R.-C.); archilla@clinic.cat (I.A.); slopezp@recerca.clinic.cat (S.L.-P.); madiaz@clinic.cat (A.D.); 2August Pi i Sunyer Biomedical Research Institute (IDIBAPS), 08036 Barcelona, Spain; mpellise@clinic.cat (M.P.); jordi.camps@uab.cat (J.C.); 3Centro de Investigación Biomédica en Red en Enfermedades Hepáticas y Digestivas (CIBEREHD), 28029 Madrid, Spain; 4Department of Clinical Foundations, University of Barcelona (UB), 08036 Barcelona, Spain; 5Pathology Department, Hospital Arnau de Vilanova, 25198 Lleida, Spain; jtarragona@gss.cat; 6Pathology Department, Instituto Valenciano de Oncología, Hospital Quirón-Salud Valencia, University of Valencia, 46010 Valencia, Spain; isidro.machado@uv.es; 7Centro de Investigación Biomédica en Red en Cancer (CIBERONC), 28029 Madrid, Spain; 8Pathology Department, Virgen de la Salud Hospital, 45071 Toledo, Spain; jmartin@sescam.jccm.es; 9Gastroenterology Department, Consorci Sanitari de Terrassa, 08227 Terrassa, Spain; dzaffalon@cst.cat; 10Gastroenterology Department, Hospital Clinic, University of Barcelona, 08036 Barcelona, Spain; daca@clinic.cat; 11Cell Biology and Medical Genetics Unit, Department of Cell Biology, Physiology and Immunology, Faculty of Medicine, Autonomous University of Barcelona (UAB), 08193 Bellaterra, Spain

**Keywords:** pT1 colorectal cancer, lymph node, staging, diagnosis, OSNA

## Abstract

**Simple Summary:**

This study focuses on very-early-stage colorectal cancer (CRC), pT1, frequently diagnosed under the umbrella of CRC screening programs. There is a great deal of debate about how best to treat pT1 CRC to avoid the overtreatment or undertreatment of patients. The authors use the RT-PCR-based quantitative molecular assay OSNA (One-Step Nucleic Acid Amplification), which detects the presence and amount of metastatic tumour cells in the lymph nodes of pT1 CRC surgical specimens. Its positivity is related to high-risk clinicopathological features. This study gives insights into the application of OSNA in early-stage CRC and its usefulness in improving management decisions.

**Abstract:**

Early-stage colorectal carcinoma (CRC)—pT1—is a therapeutic challenge and presents some histological features related to lymph node metastasis (LNM). A significant proportion of pT1 CRCs are treated surgically, resulting in a non-negligible surgical-associated mortality rate of 1.5–2%. Among these cases, approximately 6–16% exhibit LNM, but the impact on survival is unclear. Therefore, there is an unmet need to establish an objective and reliable lymph node (LN) staging method to optimise the therapeutic management of pT1 CRC patients and to avoid overtreating or undertreating them. In this multicentre study, 89 patients with pT1 CRC were included. All histological features associated with LNM were evaluated. LNs were assessed using two methods, One-Step Nucleic Acid Amplification (OSNA) and the conventional FFPE plus haematoxylin and eosin (H&E) staining. OSNA is an RT-PCR-based method for amplifying CK19 mRNA. Our aim was to assess the performance of OSNA and H&E in evaluating LNs to identify patients at risk of recurrence and to optimise their clinical management. We observed an 80.9% concordance in LN assessment using the two methods. In 9% of cases, LNs were found to be positive using H&E, and in 24.7% of cases, LNs were found to be positive using OSNA. The OSNA results are provided as the total tumour load (TTL), defined as the total tumour burden present in all the LNs of a surgical specimen. In CRC, a TTL ≥ 6000 CK19 m-RNA copies/µL is associated with poor prognosis. Three patients had TTL > 6000 copies/μL, which was associated with higher tumour budding. The discrepancies observed between the OSNA and H&E results were mostly attributed to tumour allocation bias. We concluded that LN assessment with OSNA enables the identification of pT1 CRC patients at some risk of recurrence and helps to optimise their clinical management.

## 1. Introduction

Colorectal cancer (CRC) is the third most common cancer worldwide, with 1,931,590 new diagnoses each year, and it is the second leading cause of cancer-related death in both men and women [1]. CRC carcinogenesis is related to genetic and environmental factors, with the latter comprising obesity, physical inactivity, a high red meat intake, alcohol consumption, and cigarette smoking, together with an imbalance of gut microbiota [2,3,4]. In fact, the adoption of “Western” lifestyles in some countries has led to an increase in the incidence of CRC. Thus, several countries have implemented CRC screening programmes in the average-risk population as a secondary prevention measure. CRC screening is a feasible and cost-effective intervention, since it meets all criteria for screening: (a) it has a high incidence; (b) it is perceived by the population as a health problem; (c) it has a stepwise process of development over a period of time; (d) the premalignant lesion is well known; (e) the different intermediate stages of premalignant and invasive malignant lesions are most ideal for cancer prevention and early diagnosis in the early stages when it is still curable; and (f) the screening test is inexpensive and accepted by the population. In addition, the most relevant accomplishments of CRC screening are the reduction in CRC-related deaths and the increase in early cancer detection, since about 70% of all CRCs diagnosed are at stages I-II (pT1-4 pN0), and up to 38% are pT1 CRC cases [5].

According to the published guidelines for localised CRC published by the European Society of Medical Oncology (ESMO), pT1 CRC should be treated with a complete en bloc endoscopic resection whenever possible, and patients with tumours that harbour histological high-risk factors should undergo oncologic surgery with lymph node dissection [6]. The pathological risk factors associated with lymph node metastasis (LNM) in pT1 CRC include lymphovascular invasion (LVI), perineural invasion (PNI), a high histological grade, high tumour budding (TB) and poorly differentiated clusters (PDCs), deep submucosal infiltration, and complete disruption of the muscularis mucosae [6,7,8,9]. Although a high percentage of these pT1 carcinomas are surgically treated, only 6–16% of patients have LNM, with no clear impact on survival. In addition, there is a non-negligible surgical-associated mortality of 1.5–2% [10].

Another important issue for these patients is that the detection of micrometastases in lymph nodes (LNs) may be difficult to achieve at such an early stage of the disease, since haematoxylin and eosin (H&E) sensitivity is low for the detection of small metastatic tumour nests. Therefore, despite surgical resection being performed, pN staging with H&E may not be accurate enough. For all the above, more precise methods of nodal staging are needed to determine whether surgically treated pT1 patients are at risk of recurrence.

The OSNA (One-Step Nucleic Acid Amplification) assay is a RT-PCR-based technique for detecting CK19 mRNA. It is a quantitative, fast, and standardised assay, and it has been validated for lymph node molecular analysis in breast and CRC [11,12,13]. OSNA has proven to be more sensitive than H&E for the detection of LN micrometastases in CRC [11]. The OSNA results are given as the total tumour load (TTL), defined as the sum of all copies of CK19 mRNA present in all the lymph nodes of a surgical resection specimen. Previous studies have shown that a TTL ≥ 6000 copies/µL is associated with a poorer overall and disease-free survival. Thus, TTL has been proposed as a prognostic factor in early-stage CRC [14].

The aim of this study was to analyse the LNs of surgical resections from pT1 CRCs with the OSNA assay to enable the identification of patients at risk of recurrence and the optimisation of their therapeutic management.

## 2. Materials and Methods

### 2.1. Patients and Samples

This is an observational multicentre prospective–retrospective study conducted between 2012 and 2020, including patients with a pT1 CRC who underwent surgical resection and an analysis of LNs with OSNA. Eighty-nine pT1 CRC patients were included from 4 participating centres. All patients were treated with oncologic surgical resection. Most of the recruited patients were diagnosed under the umbrella of the CRC screening programme after a positive faecal immunochemical occult blood test (FIT) (OC-Sensor^®^, Eiken, Tokyo, Japan).

A total of 40 (44.9%) patients were primarily treated with surgical resection, and 49 (55.1%) were initially treated with endoscopic resection followed by surgery, as they presented histological risk factors for LNM. Lymph nodes were assessed using two methods, H&E and the OSNA assay. The mean age was 62.6 years old (range 31–88), and 49 (55.1%) were male. Most tumours were on the left colon (44; 49.4%) and originated on non-pedunculated polyps (71; 79.8%). Relevant clinicopathological data are presented in Table 1.

### 2.2. Ethical Considerations

This study was approved by the Ethics and Scientific Committee of our institution (HCB 2012/7324). All patients signed and kept a copy of the informed consent document for participation in the study after the nature of the research was fully explained. Another copy was kept with each patient’s clinical files, and a third copy was kept in the Biobank files of our institution.

### 2.3. Lymph Node Dissection and One-Step Nucleic Acid Amplification (OSNA)

Lymph nodes from surgical specimens were freshly dissected within 45 min after surgery, using the pooling method as previously described by Rakislova et al. [15], which analyses multiple lymph nodes in the same PCR tube, up to a 600 mg weight limit of the OSNA assay. Briefly, we used one-half of the LN for conventional processing, with formalin fixation and paraffin embedding (FFPE). Then, 3 µm thick cuts of paraffin blocks containing the LNs were stained with H&E. The other half of the LN was used for an OSNA analysis.

The OSNA molecular technique (Sysmex Corporation, Kobe, Japan) is a quantitative assay, which amplifies cytokeratin 19 (CK19) mRNA from lymph node lysates in three steps, the homogenisation, centrifugation, and amplification of CK19 mRNA using RT-LAMP (Reverse Transcription Loop-mediated Isothermal Amplification). Firstly, the LN tissue is homogenised with 4 mL of Lynorhag lysate buffer for 60 s using RP-10 equipment (Sysmex, Kobe, Japan), which stabilises the mRNA molecules and protects them against ribonuclease activity, minimising the effect of inhibitory substances and precipitating the genomic DNA. Subsequently, 1 mL of the resulting product is centrifugated at 12,000× *g* for 1 min at room temperature to remove cell debris. Then, amounts of 500 + 500 μL are taken from the intermediate phase and put into two PCR tubes, corresponding to the backup sample and the main sample. Then, 20 μL from the main sample is diluted in 180 mL of Lynorhag buffer to a 1:10 concentration (analysis sample), which is put into an RD-210 amplifier (Sysmex Europe GmbH), and the isothermal amplification of CK19 mRNA is performed at 65 °C using the Lynoamp reagent. RD-210 can analyse 14 samples simultaneously. The backup sample is frozen at −80 °C for reanalysis if needed. During the amplification reaction, pyrophosphate is produced, which binds to a magnesium ion, turning into magnesium pyrophosphate, which has a low solubility in aqueous solutions and precipitates when its concentration reaches saturation, producing turbidity in the medium, which is detected using the RD-210 platform. Once the turbidity reaches an optical density value of 0.1 at 465 nm, a time threshold is established, from which the CK19 mRNA copy number is determined. The turbidity measurement is performed in real time at 6 s intervals.

### 2.4. Lymph Node pN Staging and OSNA Results

For all cases, pN staging was reported using conventional H&E staining of part of the LN used for FFPE.

The OSNA results were blind to both the pathologists and the clinicians, and they were obtained from the analysis of the other part of the LN. The OSNA results are expressed as the TTL, defined earlier. A TTL of ≥250 copies of CK19 m-RNA/µL was defined as the positive threshold of the technique. The cut-off value that we considered as clinically significant was ≥6000 copies/µL, as this has been reported to have prognostic value [14].

### 2.5. Histological Evaluation of pT1 CRC

The H&E-stained slides of all pT1 CRCs were re-evaluated by a gastrointestinal pathologist (MR) and a fellow resident (KS), following the published guidelines for pT1 pathological diagnosis, without any information about the patient’s follow-up or lymph node status [8,16,17,18,19,20,21,22]. The only clinical information available at the time of the histological evaluation was the endoscopy report assessing the location and type of polyp (pedunculated or non-pedunculated). A double-head Olympus BX51 microscope (Olympus, Tokyo, Japan) was used for the evaluation of all histological factors, i.e., polyp size, tumour size, histological grade, TB, PDC, LVI, PNI, the disruption of the muscularis mucosae, the depth of submucosal infiltration, and resection margin status.

### 2.6. Statistical Analysis

Fisher’s exact test was used to measure the association between lymph node status and the frequencies of categorical variables, whereas the Kruskal–Wallis test was used to compare the means and medians of quantitative variables and LN status. *p* < 0.05 was considered statistically significant. The concordance of H&E and OSNA was evaluated with the percentage of agreement. For this analysis, we used the R program (4.0.3 version; R Foundation for Statistical Computing, Vienna, Austria).

## 3. Results

### 3.1. Comparison of Lymph Node Status Assessed with H&E and OSNA

All lymph nodes were assessed with both H&E and OSNA, using half of the LNs for each method. In eight cases (9.0%), the lymph nodes resulted positive with H&E, and 22 cases (24.7%) resulted positive with OSNA. In three cases, the TTL was >6000 copies/μL.

In 72 cases (80.9%), there was agreement in the results of the two methods, with 6 cases being concordantly positive and 66 cases being concordantly negative. One of the latter cases had 10 freshly dissected LNs determined to be concordantly negative with both H&E and OSNA. In addition, 13 extra LNs were dissected after formalin fixation, and they were only analysed with H&E; 1/13 positive LNs resulted in pN1a staging. The H&E-positive cases (pN1) had a mean TTL of 9241 copies/μL (median 5000 copies/μL; range 220–40,740 copies/μL).

Among the 17 discordant cases, 16 were found to be positive with OSNA and negative with H&E, with all of them harbouring TTL < 6000 copies/μL (mean 1540.6 copies/μL; median 990 copies/μL; range 260–5220 copies/μL). One case was found to be positive with H&E and negative with OSNA, and it had a TTL of 220 copies/μL (below the 250 copies/μL threshold of the method for positivity). This case can be considered a false negative of the OSNA technique, probably due to tumour allocation bias (Figure 1).

### 3.2. Pathological Characteristics Related to Lymph Node Metastases

The histological features associated with LNM when assessed with H&E were TB (*p* = 0.015), PDC (*p* = 0.028), and the presence of LVI (*p* = 0.03). On the contrary, no association was found with PNI, histological grade, the disruption of the muscularis mucosae, the status of the resection margins, tumour size, or the depth of submucosal infiltration.

We then assessed the association of the histological features with the OSNA results, comparing the cases below and above TTL 6000 copies/µL, since this value is related to prognosis. Although there were only three cases with a TTL > 6000 copies/µL, a statistically significant association was observed with TB (*p* = 0.038). A non-significant trend was observed with PDC and LVI (*p* = 0.073 each). All other histological factors showed no association (Table 2 and Figure 2).

### 3.3. Patient Follow-Up

Six patients determined to have positive lymph nodes with H&E received adjuvant chemotherapy. The follow-up period was between five months and eight years, and most patients were alive without disease (82 patients; 92.1%). Three patients died from other causes, and four were lost to follow-up. None of the patients had distant metastases or recurrence of the disease, and no differences regarding the prognosis of the patients were found when the LN status was assessed with either H&E or OSNA (*p* = 0.62 and *p* = 0.58, respectively).

## 4. Discussion

This study demonstrates that an analysis of the LNs from pT1 CRCs with the OSNA quantitative molecular assay, which detects the amount of tumour burden in the LNs, can identify patients at some risk of progression. This highly sensitive molecular method of LN analysis enables the detection of pT1 CRCs capable of LN tumour cell seeding, and it separates them from those with completely negative LNs; therefore, it enables the optimisation of pT1 CRC patients’ therapeutic management. Moreover, the quantitative TTL results are associated with the clinicopathological features related to LNM. The assessment of the amount of tumour cells within the LNs gives additional prognostic information on early-stage CRC, considering that a TTL of ≥6000 copies/µL has been associated with a poorer overall- and disease-free survival [14]. Therefore, in this study, we only considered a TTL of ≥6000 copies/µL with potential clinical relevance. Initially, LN lymph node micrometastases in CRC were thought to have no clinical significance [23], but a meta-analysis by Rahbari et al. revealed that molecular tumour cell detection in regional LNs, otherwise found to be negative with H&E (pN0), was associated with poor overall survival, disease-free survival, and disease-specific survival [24,25]. Interestingly, another meta-analysis by Sloothaak et al., which included studies that had used immunohistochemistry to assess LNM, reported that patients with micrometastases were at higher risk of tumour recurrence. However, the finding of isolated tumour cells (ITCs) had no impact on disease-free survival [26].

A relevant finding of our study was the higher LN positivity found when using OSNA (22 cases; 24.7%), compared to the positivity found when using H&E (7 cases; 7.9%). The three cases with TTL > 6000 copies/µL were also positive with H&E, which reflects the importance and reliability of TTL values. There were sixteen cases with negative LNs with H&E but positive with OSNA, with TTL < 6000 copies/µL, which may represent either micrometastases or ITC. The discordances are most likely explained by tumour allocation bias, as part of the LN was used for OSNA and part was used for conventional staging with H&E, which analyses only one 3 µm central section of the LN [27,28].

We analysed the histological features associated with LNM with LN positivity using both H&E and OSNA. In line with the results reported by Archilla et al. [14], we also observed that TB was associated with OSNA TTL ≥ 6000 copies/µL and H&E positivity. Tumour budding is a proven risk factor for LNM in pT1 CRC [7,16,18,29,30,31,32,33,34,35,36,37,38,39,40], and the current guidelines recommend undergoing surgery in cases with high TB (Bd3) [6]. PDCs have been described more recently, and they are defined as cancer nests composed of ≥5 cancer cells showing no gland-like structure [16,17,18,19]. PDCs are considered an histological risk factor for LNM in pT1 CRC [9,41]. We found that PDCs were associated with LNM assessed with H&E, with a non-significant trend with OSNA, probably due to the small number of cases with a TTL > 6000 copies/µL. Both TB and PDC have been proposed to be evaluated together in a “Combined Score”, as they both may refer to the same biological phenomenon of epithelial-to-mesenchymal transition, but there is still some controversy among experts [42,43,44,45]. In addition, we also observed that the presence of LVI was associated with LNM when assessed with H&E. As is well known, the presence of LVI is a significant risk factor for LNM in pT1 CRC [5,7,8,9,16,17,18,29,30,35,36,37,38,39,40,41,46,47,48].

We did not find an association between PNI and LNM with either method of LN analysis. However, we must consider that PNI is a feature seldom found in pT1 CRC. Nevertheless, an association between LNM and PNI has been widely demonstrated in more advanced stages of CRC, which is also related to disease-free and overall survival [7,10,16,24,26,29,31,37,38,45,47,49]. Further, we found no relation between LNM and the depth of submucosal infiltration when analysed with either method. This feature has been associated with LNM in several studies, but different cut-off points have been considered for risk stratification [7,8,17,18,29,30,31,32,34,35,36,37,38,40,47,50]. In a recent meta-analysis, Zwager et al. showed no significant association between LNM and the depth of submucosal infiltration [32]. A high histological grade is also considered to confer a higher risk for the presence of LNM, being consistently reported in the literature [5,7,8,16,29,30,32,34,35,36,37,38,40,45,46,47,51,52], while other studies have not found such a relationship [9,17,39,41,48]. In our study, we did not observe any association between a high histological grade and LNM. Finally, another histological feature that we found not to be related to LNM was the disruption of the muscularis mucosae. There has been some debate about the importance of the complete disruption of the muscularis mucosae. While some authors have demonstrated its association with LNM, not only in pedunculated polyps but in all types of polyps, other authors have not observed such a link [9,30,31,36].

Although a histopathologic analysis remains the gold standard in the diagnosis and risk classification of CRC, it has many limitations. In recent years, the introduction of artificial intelligence (AI) has influenced different fields of medicine, including CRC diagnosis [53]. AI may be used as a diagnostic tool for tissue classification and the prediction of outcomes [54,55]. In pT1 CRC, AI models identify patients at risk of LNM more accurately than histopathologic diagnosis following the current guidelines [56].

The meaning of the amount of TTL is of outmost importance, and, therefore, it is in accordance with its clinical significance. In fact, the TTL of sentinel lymph nodes (SLNs) in breast carcinoma is associated with disease-free survival and overall survival, and, in the intraoperative setting, the TTL of SLNs is a surrogate indicator of the status of non-SLNs; thus, its value is critical for deciding the need to perform lymphadenectomy [57,58]. In this same context, the role of the molecular detection of LN metastasis using OSNA in patients with breast cancer who have received systemic neoadjuvant therapy has recently been explored. Thus, a TTL ≥ 15,000 copies/μL in SLNs has been shown to be a predictive factor for metastasis in non-sentinel lymph nodes, with a negative predictive value (NPV) of 90.5%, while a TTL ≥ 25,000 copies/μL is related to worse disease-free survival [59].

Before the demonstration of the prognostic value of the TTL in CRC, there was some evidence indicating that low TTL values of ≥250 copies/µL, although considered “positive”, did not correspond to patient upstaging, and were not associated with prognosis [60]. The reported rates of positivity and of nodal upstaging using OSNA in CRC are variable among different studies; i.e., Aldecoa et al. reported that up to 51% of pN0 patients with H&E had OSNA-positive LNs, while Croner et al. found it to be 25.2%, Yamamoto et al. found it to be 17.6%, and Tani et al. found it to be 9.1% [61,62]. Some authors initially observed an association between the TTL with high-risk clinicopathological features and observed that pT1 CRC had lower TTL values than tumours with deeper infiltration, finding a correlation between the TTL and pT stage [61,62]. In line with this, Yamamoto et al. showed that the TTL was related to pN stage with H&E, as pN0 cases had a median of 1500 copies/µL, while those with pN1 had 24,050 copies/µL, and pN2 cases had 90,600 copies/µL [27]. Archilla et al. found similar results, with a mean TTL in pN0 of 1775 copies/µL, pN1 of 49,413 copies/µL, and pN2 of 95,000 copies/µL [14]. Their results were validated by Diaz-Mercedes et al., observing a mean TTL in pN0 of 433.6 copies/µL, pN1 of 35,782.6 copies/µL, and pN2 of 144,651.8 copies/µL [14,27,63]. In our study, the tumour burden in the LNs of pT1 CRC with pN0 determined with H&E was lower, with a median of 990 copies/μL (mean 1540.6 copies/μL), while pN1 cases had a median of 5000 copies/μL (mean 9241 copies/μL).

Therefore, an important message about LN status when detected with highly sensitive molecular methods is that any OSNA positivity is not equivalent to patient upstaging and that low TTL values may not have clinical consequences. In fact, the high sensitivity of OSNA can detect tumour cell seeding to LNs, even in pTis CRC, as demonstrated by Rodrigo-Calvo et al., who found that TTL values below 6000 copies/µL (range 400 to 4270 copies/µL) had no clinical significance. They suggested that these TTL values might represent the presence of ITCs in the LNs of pTis CRC [64]. Yet, the meaning of the presence of ITC in CRC is still controversial. In view of their results, it would be of interest to reliably distinguish the significance of ITC from micrometastases assessed with OSNA.

Our study has some limitations. The most determinant one is that part of the LNs was assessed with H&E and part was assessed with OSNA, which causes an intrinsic tumour allocation bias whenever the tumour metastasis is little enough to be contained only in one part of the LN. In fact, this is the most plausible explanation for our discordant cases. This could be solved by submitting the whole LN for molecular analysis, as carried out in LNs from breast carcinoma [13,28]. In fact, all the OSNA studies on CRC to date have used similar methodologies, with part of the LN used for H&E and part used for OSNA, which may be the reason for the false negativities and the limiting of its applicability in general practice [65]. We had 1.1% false negatives, but this rate has been reported up to 15.4% in other series [61]. Despite this limitation, almost all published studies have shown good performance of this technique, showing a high sensitivity (90.4%), specificity (96.8%), concordance (96.0%), positive predictive value (79.8%), and negative predictive value (98.6%) [28,65]. In this regard, OSNA LN staging can be improved and achieve its full clinical significance and impact in early-stage CRC only through a complete analysis of the LN tissue. Only then will the TTL values be able to be validated for CRC LN assessment. In this regard, Diaz-Mercedes et al. performed an intermediate step by demonstrating that LN cytological smears could be used to determine the pN stage, enabling the use of the whole LN tissue for OSNA. In fact, they found that cytological LN smears immunostained with CK19 had a similar performance to the OSNA assay in the determination of LN positivity, and they were more sensitive than H&E (*p* < 0.0001) [63]. Another limitation of our study is the small number of cases, since the recurrence of disease in pT1 CRC is only 3.3%, and the proportions of cases with metastases and death from the disease are even lower, being 1.6% and 1.7%, respectively [40]. Therefore, we are unable to make any statement about the prognostic implications of the LN status assessed using either OSNA or H&E. Another minor issue is the finding of LNs located in the fat very close to the colorectal wall, which may not be freshly dissected, and they were therefore assessed only using H&E. Indeed, one of our cases had LNM in an additional LN dissected after formalin fixation. Missing lymph nodes in fresh dissection can occur, but all LNs dissected after FFPE must be assessed only using conventional H&E, and they still contribute to the pN stage [62,64].

To the best of our best knowledge, this is the first OSNA study focusing on pT1 CRC. We have demonstrated that it enables the identification of patients at risk of recurrence, which helps the optimisation of their clinical management.

## 5. Conclusions

A molecular analysis of LNs from pT1 CRCs provides more information on the real nodal status than H&E. Patients with TTL ≥ 6000 copies/µL may be candidates for a closer follow-up and could be considered for adjuvant therapy; therefore, an OSNA LN analysis can contribute to improve the management of patients with early-stage CRC.

## Figures and Tables

**Figure 1 cancers-15-05481-f001:**
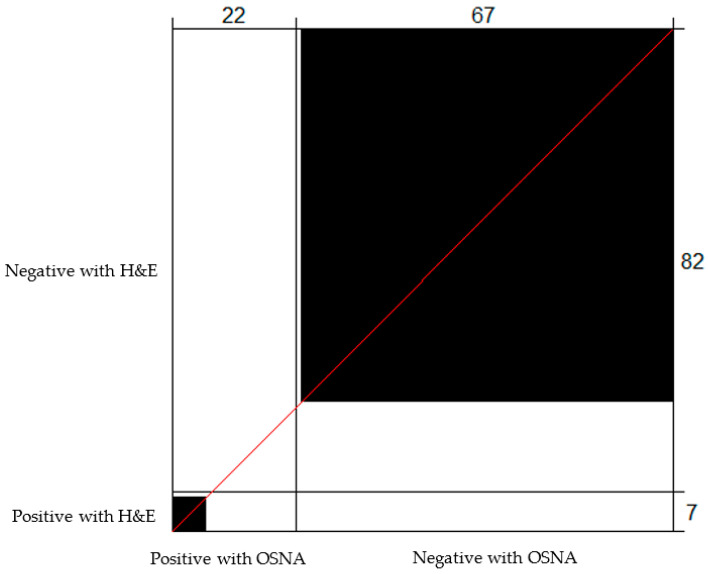
Bangdiwala’s Observer Agreement Chart comparing the lymph node status determined using OSNA and H&E. The black boxes correspond to concordant cases. Of the 22 cases determined to be positive with OSNA, H&E was positive for 6, and 66 out of 82 cases determined to be negative with H&E were also determined to be negative with OSNA.

**Figure 2 cancers-15-05481-f002:**
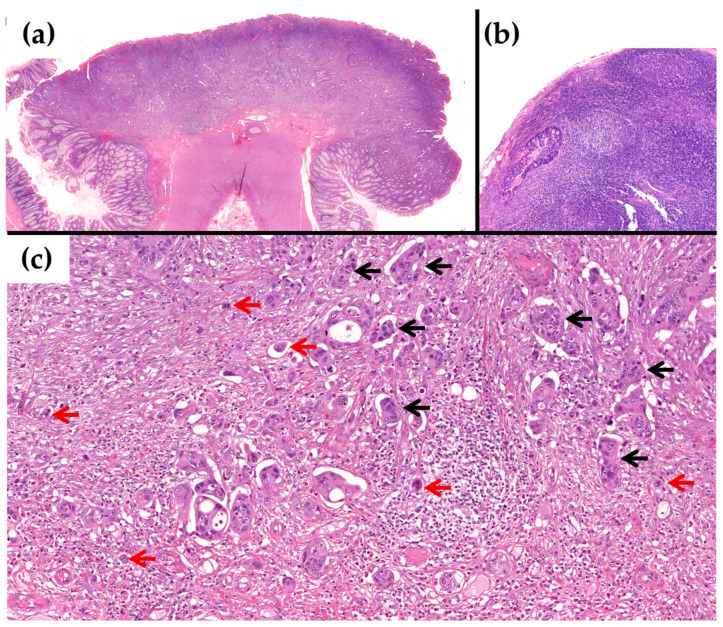
Example of a case with TTL > 6000 copies/µL with OSNA and a positive LN status with H&E. (**a**) Low power of the pT1 CRC with submucosal infiltration, which was resected primarily on surgery (H&E × 20). (**b**) Lymph node with metastasis of the adenocarcinoma (H&E × 100). (**c**) At higher power, the tumour shows high TB, Bd3 (red arrows), and G3 PDC (black arrows) on the infiltrating margin (H&E × 200).

**Table 1 cancers-15-05481-t001:** Clinical and pathological features of pT1 colorectal carcinomas.

Category	N (Total)	N (%)
Gender		
Male	49	55.06
Female	40	44.94
Location		
Right colon	33	37.08
Left colon	44	49.44
Rectum	12	13.48
Polyp type		
Pedunculated	18	20.22
Non-pedunculated	71	79.78
Follow-up		
Alive without disease	82	92.13
Dead from other causes	3	3.37
Lost to follow-up	4	4.49

**Table 2 cancers-15-05481-t002:** Pathological risk factors associated with LN status assessed with H&E and OSNA.

	Total (%)	H&E Negative (%)	H&E Positive (%)	*p*-Value	OSNA 250-5999 Copies/μL (%)	OSNA ≥ 6000 Copies/μL (%)	*p*-Value
Histological grade				0.093			0.470
Low grade	72 (80.9)	66 (91.7)	6 (8.3)		16 (88.9)	2 (11.1)	
High grade	17 (19.1)	13 (76.5)	4 (23.5)		3 (75.0)	1 (25.0)	
Tumour budding				0.015			0.038
Bd1	73 (82.0)	68 (93.2)	5 (6.8)		18 (94.7)	1 (5.3)	
Bd2	10 (11.2)	7 (70)	3 (30)		1 (33.3)	2 (66.7)	
Bd3	6 (6.7)	4 (66.7)	2 (33.3)		0 (0)	0(0)	
Poorly differentiated clusters				0.028			0.073
G1	77 (86.5)	71 (92.2)	6 (7.8)		17 (94.4)	1 (5.6)	
G2	8 (9.0)	5 (62.5)	3 (37.5)		2 (50.0)	2 (50.0)	
G3	4 (4.5)	3 (75)	1 (25)		0 (0)	0 (0)	
Disruption of muscularis mucosa				1			0.470
Partial	30 (33.7)	27 (90.0)	3 (30.0)		3 (75.0)	1 (25.0)	
Complete	59 (66.3)	52 (88.1)	7 (11.9)		16 (88.9)	2 (11.1)	
Lymphovascular invasion				0.003			0.073
Absent	77 (86.5)	72 (93.5)	5 (6.5)		17 (94.4)	1 (1.3)	
Present	12 (13.5)	7 (58.3)	5 (41.7)		2 (16.7)	2 (16.7)	
Perineural invasion				0.112			0.1364
Absent	88 (98.9)	79 (89.8)	9 (10.2)		19 (21.6)	2 (2.3)	
Present	1 (1.1)	0 (0)	1 (100)		0 (0)	1 (100)	
Resection margin status				0.804			1
Free	74 (83.1)	66 (89.2)	8 (10.8)		17 (23.0)	3 (4.1)	
Affected	9 (10.1)	8 (88.9)	1 (11.1)		1 (11.1)	0 (0)	
Non assessable	6 (6.7)	5 (83.3)	1 (16.7)		1 (16.7)	0 (0)	
Tumour size (mm)	Median	10	12.5	0.412	12	10	0.502
Depth of submucosal infiltration (mm)	Median	2.5	3	0.521	2.5	3	0.438

## Data Availability

The datasets generated and/or analysed in the current study are available from the corresponding author on reasonable request.

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
