# Peer review of "Lymph Node Molecular Analysis with OSNA Enables the Identification of pT1 CRC Patients at Risk of Recurrence: A Multicentre Study"

_cancers, 2023, doi:10.3390/cancers15225481_

Round 1
Reviewer 1 Report
Comments and Suggestions for Authors
I have no additional questions for the authors
Author Response
Please see the document attached.

Reviewer 2 Report
Comments and Suggestions for Authors
Dear Authors
Manuscript describes a novel study on “Lymph node molecular analysis with OSNA enables to identify pT1 CRC patients at risk of recurrence”.
The RT-PCR-based quantitative molecular assay OSNA (One-Step Nucleic Acid Amplification), which detects the presence and amount of metastatic tumour cells in the lymph nodes of pT1 CRC surgical specimens. Its positivity is related to high-risk clinicopathological features. This study gives insights into the application of OSNA in early-stage CRC and its usefulness for improving management decisions.
Minor correction is required,
Materials and methods – Please add how many samples and Table 1 in the 2.1 patients and samples section delete section 3.1.
Moderate editing of English language required.
Author Response
Please see the document attached.

Reviewer 3 Report
Comments and Suggestions for Authors
This manuscript introduces and comprehensively explores the role of a lymph node (LN) staging method to optimize CRC pT1 patients’ therapeutical management in order not to overtreat or undertreat them
The topic is original and relevant to the field. There is limited information on this topic in the literature.
There are no further improvements regarding the methodology.
The conclusions are consistent with the evidence and arguments presented as well as summarize the main point of this article.
References are up-to-date and appropriate
Figures and tables are well formatted and make the study easy to follow
Minor revision
1) "Colorectal cancer is one of the most prevalent types of cancer, with histopathologic examination of biopsied tissue samples remaining the gold standard for diagnosis. During the past years, artificial intelligence (AI) has steadily found its way into the field of medicine and pathology, especially with the introduction of whole slide imaging (WSI)."
I would suggest adding this important information to the discussion section and a brief discussion on Tissue classification and diagnosis of colorectal cancer histopathology images using deep learning algorithms
https://www.termedia.pl/Tissue-classification-and-diagnosis-of-colorectal-cancer-histopathology-images-using-deep-learning-algorithms-Is-the-time-ripe-for-clinical-practice-implementation-,41,51207,0,1.html
2) I would suggest a brief discussion on The role of deep learning in diagnosing colorectal cancer
Consider citing the recently published article
https://www.termedia.pl/The-role-of-deep-learning-in-diagnosing-colorectal-cancer,41,51103,0,1.html
Author Response
Please see the document attached.
